# Cohesive Conversations: Enhancing Authenticity in Multi-Agent Simulated Dialogues

**KuanChao Chu, Yi-Pei Chen & Hideki Nakayama** [*]
The University of Tokyo
{kcchu,ypc}@nlab.ci.i.u-tokyo.ac.jp

## Abstract

This paper investigates the quality of multi-agent dialogues in simulations powered by Large Language Models (LLMs). Analyzing dialogues and memory over multiple sessions revealed significant issues such as repetition, inconsistency, and hallucination, exacerbated by the propagation of erroneous information. To combat these challenges, we propose a novel Screening, Diagnosis, and Regeneration (SDR) framework that detects and corrects utterance errors through a comprehensive process involving immediate issue identification, evidence gathering from past dialogues, and LLM analysis for utterance revision. By incorporating our SDR framework to Generative Agents (Park et al., 2023), we enhance the diversity, consistency, and factualness of the generated dialogues. This work presents a pioneering approach to enhancing dialogue quality in multi-agent simulations, establishing a new standard for future research in the field.

## 1 Introduction

Recent research has leveraged Large Language Models (LLMs) (OpenAI, 2023; Touvron et al., 2023) to power multi-agent simulations, aiming to model complex human behaviors (Park et al., 2023) or enhance multi-agent communication and collaboration (Chen et al., 2023b; Hong et al., 2023). In these simulations, agents are equipped with an LLM as their core, augmented with additional components such as memory, and communicating iteratively (Cheng et al., 2024b). Multi-agent simulated dialogues can be adapted for entertainment purposes such as non-player characters (NPCs) in video games for more dynamic and adaptable interactions (Replica Inc., 2023; AI Dungeon, 2024). They can also facilitate more effective decision-making through simulating human interactions (Chen et al., 2023b; Wang et al., 2024), or creating dialogue scripts in movies and novels (Chen et al., 2023c; Maas et al., 2023; Cheng et al., 2024a). It is essential to ensure the sustained quality of interactions among multiple agents in long-term simulations.

Previous works evaluate multi-agent simulation by task success rate in goal-oriented tasks (Hong et al., 2023; Wang et al., 2023a; Chen et al., 2023b) or through "interviewing" the agents (Park et al., 2023). The evaluation of long-term (multi-session) dialogues is limited to sessions involving the same pair of speakers (Xu et al., 2022; Bae et al., 2022; Jang et al., 2023). None of the previous research has thoroughly scrutinized the multi-agent communication over time, leaving a critical area of research unaddressed.

Our work delves into the simulation log of Park et al. (2023)[1] (hereafter referred to as ONEDAYLIFE), which simulates 25 agents "living" in a village over a day and sometimes conversing with each other.[2] Examining the dialogue content and memory of the participating agents across multiple sessions, we found that while individual dialogues may appear flawless on the surface, a holistic review of multi-session multi-agent dialogues identifies significant issues. Specifically, we identified three main problems: repetition, inconsistency,

---

[*]The first two authors contributed equally.

[1]https://reverie.herokuapp.com/arXiv_Demo/

[2]Please refer to Appendix A and the original paper for more details about the simulation.

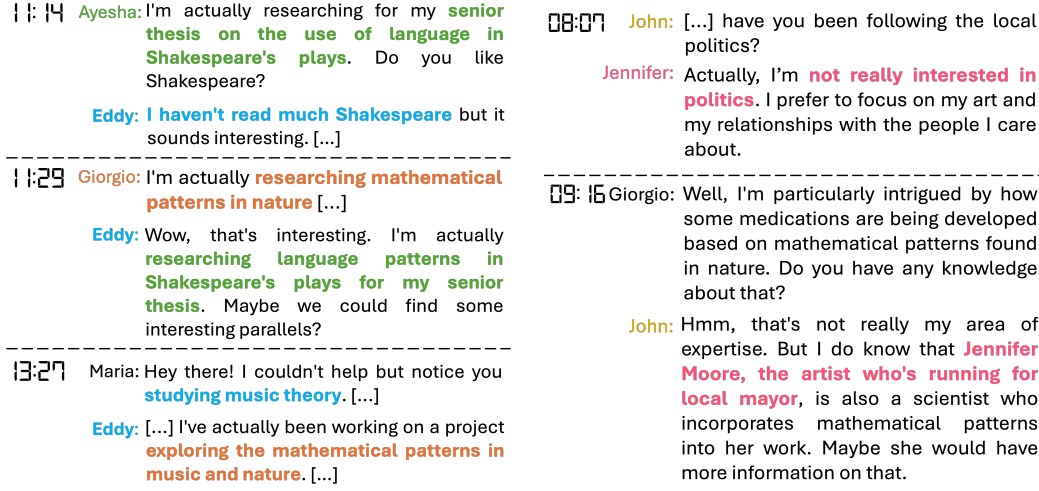

(a) Eddy is a music-major student working on a music composition project, yet he is inconsistent with what he is working on and repeating other's utterances after conversing with them.

(b) Despite Jennifer telling John that she is not interested in politics, John hallucinates that Jennifer is running for local mayor election in the dialogue with Giorgio.

Figure 1: Example dialogues from ONEDAYLIFE showing problems of repetition, inconsistency, and hallucination. Each agent name is colored, and the bold-colored phrases indicate the mentioned attribute's owner or the original speaker.

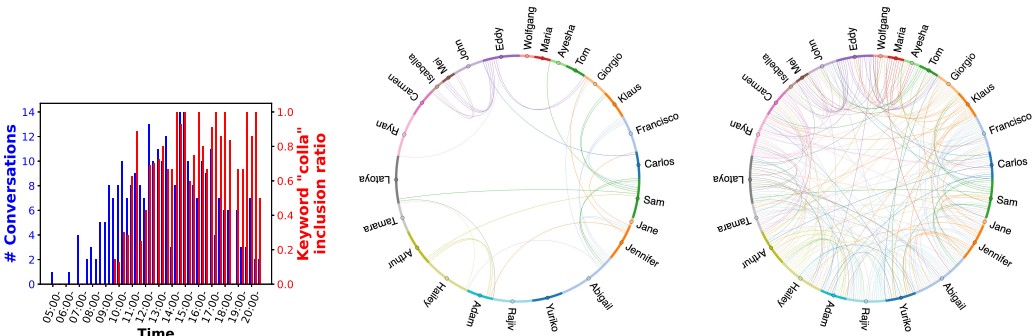

Figure 2: The spread of the keyword "collaboration" in ONEDAYLIFE. **Left:** The number of dialogues and the ratio that includes the keyword in each time span. **Middle:** Number of dialogues with the keyword in the first 20% of spreading time. Each line represents a dialogue between two agents and the line color indicates the identity of the agent who firstly mentions the keyword. **Right:** Number of dialogues with the keyword in all time.

and hallucination, as depicted in Fig. 1 (further examples are available in Appendix D.3). Notably, these problems escalate with the duration of the simulation, exacerbated by the propagation of erroneous information through dialogues among agents. Although human conversations sometimes also include repetitions, inconsistencies, and false statements, the frequency of such behavior should not be as high as presented in the LLM-synthesized dialogues. For example, all agents "collaborate" with others at the end of the simulation, as shown in Figure 2.

To address these challenges, we introduce a novel multi-agent simulation framework featuring a Screening, Diagnosis, and Regeneration (SDR) mechanism. This approach enables the immediate detection and correction of errors in utterances generated by agents. During the Screening stage, potential issues in the candidate utterance are identified, and relevant evidence from past dialogues is gathered. Subsequently, in the Diagnosis stage, an LLM

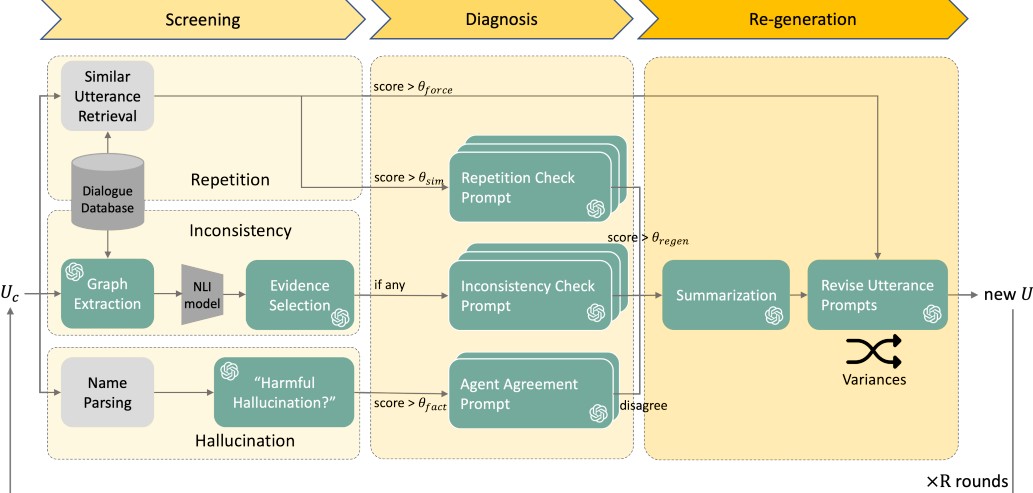

Figure 3: Overview of the proposed Screening, Diagnosis, Re-generation (SDR) framework, an instant error correction method for multi-agent simulated dialogues. The modules in green are run by the LLM.

analyzes the evidence and the current dialogue, providing detailed feedback and a score on the authenticity of the candidate utterance. If it is deemed problematic, it is revised during the Regeneration stage, incorporating insights from the preceding stages. Both GPT-4 assessments and human evaluations validate that our multi-agent, multi-session dialogues are more consistent and have fewer false hallucinations, while automatic metrics confirm that we achieve better diversity, confirming the effectiveness of our SDR framework.

In summary, our contributions are threefold: (1) We illuminate the inherent problems within multi-agent, multi-session dialogues. To the best of our knowledge, we are among the first to investigate this particular problem. (2) We propose the SDR framework, a pioneering approach for on-the-fly detection and correction of utterance errors in multi-agent simulations. (3) Through extensive evaluations and analyses, we demonstrate the efficacy of our framework, setting a new standard for dialogue quality in multi-agent simulations.

## 2 Multi-agent Dialogue Simulation

We propose a Screening, Diagnosis, and Regeneration (SDR) framework for simulating authentic dialogues between multiple agents over a period of time. For each candidate utterance $U_c$ generated by an agent, we examine whether potential errors occur, and re-generate a new $U_c$ if any problem is found. Figure 3 illustrates our SDR system overview.

### 2.1 Screening

Due to the limitations of LLMs, including input length restrictions and associated costs, it is impractical to compare each $U_c$ against all past dialogues. Consequently, we initiate a Screening stage to identify potential issues and gather pertinent evidence for each identified issue.

**Repetition** Agents often display similar speech patterns, reducing their character distinctiveness. As shown in Fig. 1a, Eddy tends to replicate phrases from other agents following their conversations. Figure 2 illustrates that as the day progresses, most agents engage in highly similar topics related to collaboration. More examples can be found in Appendix D.3.

We build a dialogue database that stores all utterances prior to the candidate utterance $U_c$, which includes utterances from previous dialogues and the current dialogue context.

For each $U_c$, we first query the database to retrieve the top $K_{sim}$ similar utterances. We apply a dynamic similarity threshold $\theta_{sim}$ to identify the excessively repetitive utterances, determined by whether they originated from the same speaker or the same dialogue or not. More details are explained in Appendix B.1. If more than one retrieved utterances surpass $\theta_{sim}$, all dialogues associated with these utterances will be marked as evidence for the next diagnosis stage. For the cases when $U_c$ is nearly identical to a retrieved utterance (the similarity greater than $\theta_{force}$), we directly bypass all pipelines and proceed to the Re-generation stage.

**Inconsistency** Factual or logical inconsistencies are an issue across multiple dialogues. For instance, Fig. 1b illustrates how John's statement contradicts Jennifer's earlier words. Other examples include sudden shifts in opinions, forgetting past statements, and invitations to conflict.

We propose a Natural Language Inference-Graph (NLI-G) module for inconsistency screening. NLI-G consists of three steps. First, we employ the LLM to extract personal information as a list of (Subject, Relation, Object) triplets from each previous dialogue of involved agents, as well as from the candidate utterance $U_c$. For example, [["Giorgio Rossi", "working on", "mathematical patterns in nature"], ["Eddy Lin", "researching", "language patterns in Shakespeare's plays"]] are extracted from the dialogue between Giorgio and Eddy at 11:29 in Fig. 1a. Refer to Appendix B.2 for the results of a full dialogue. After transforming triplets into text form, we adapt an NLI model to predict potential contradictions by comparing those from previous utterances with those from $U_c$. Utilizing such a graph format helps the NLI model to focus on key information of agents and reduce the negative impact of style discrepancies between pretrained data and raw dialogue utterances. Finally, the triplets whose contradiction score is above $\theta_{nlig}$ are considered suspicious and forwarded to the LLM to select top $K_{nlig}$ corresponding dialogues for the next Diagnose stage.

**Hallucination** We focus on detecting the harmful hallucination, which we refer to as the generated spurious information in $U_c$ that is related to other agents. Such hallucination can persist within the simulated world through memories, and sometimes become truth over time, despite not aligning with the involved agent's role. Given that agents are considered distinct individuals, preventing fabricated information about others is crucial.

We detect third-party agent mentions via name parsing. Upon identification, we prompt the LLM to rate the utterance on a 1 to 10 scale, indicating the likelihood of being a hallucination, and we flag $U_c$ if the score is greater than $\theta_{fact}$. Two example cases are shown in Fig. 4.

The model is explicitly guided to give a higher score if the following criteria are met: (a) Speaker-objectivity: if it is an objective statement regarding another agent from the speaker's perspective. (b) Discernment capability of the mentioned agent: if the referenced agent can currently verify the statement's truthfulness. (c) Impact: the statement, if fabricated but later accepted as truth, significantly impacts the agent.

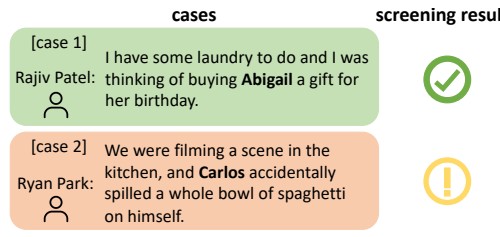

Figure 4: Examples of hallucination screening. In Case 1, although Abigail is mentioned, it pertains only to Rajiv's personal plan, not to a fact about Abigail. In Case 2, Ryan objectively describes a past event involving Carlos. However, this event could have been entirely fabricated by Ryan, representing a potential harmful hallucination.

## 2.2 Diagnosis

We utilize the LLM to further diagnose the authenticity of $U_c$ if there are evidence dialogues provided or if $U_c$ is flagged from the previous Screening stage. In the former case, we prompt LLM with pipeline-specific prompts given the evidence and current dialogues. The outputs are a score representing the severity of the issue and the reason for the output score.

In the latter case, we provide the information of the mentioned agent and ask the LLM to output whether the mentioned agent agrees with $U_c$, along with an explanatory comment. In practice, we repeat the Diagnose stage for $N_{diag}$ trials and select the one with the highest score. The prompts are provided in Appendix E.

### 2.3 Re-generation

We collect results from all pipelines and retain only comments with a score above $\theta_{regen}$ or those indicating disagreement. If no comments remain, the correction process terminates, $U_c$ is saved to the dialogue database, and the model continues to generate the next utterance of the other agent. Otherwise, the LLM is used to integrate all comments and provide suggestions for improvement, alleviating the vagueness from simple feedback (Liang et al., 2023). The prompt for re-generation is enriched by appending comments to the original prompt that was used to generate $U_c$.

Our SDR procedure continues until either of the conditions is met: completes $R$ rounds of iteration, or reaches a point where no further comments are provided, indicating the resolution of identified issues.

### 2.4 Prompt Design

We develop multiple prompt variants to increase diversity when re-generating the revised $U_c$, inspired by the finding that varied linguistic prompts induce output variance (Leidinger et al., 2023). The LLM often re-generates an exactly identical or very similar response given that most of the prompt content is the same as the initial response generation prompt, despite of providing additional feedback to guide the generation and setting the penalty for frequency and presence [3]. We design two types of utterance generation prompts, a persona-based narrative prompt and a structured task-oriented prompt. The former prompt asks the model to play the role of the given persona and engage in a conversation, which is more narrative and immersive. The latter breaks down all information into clear components, which is less about storytelling and more about providing structured data for a specific task (in this case, generating a response in a conversation). The prompts of these two types are shown in Appendix E.4.

## 3 Experiment

### 3.1 Data and Settings

We conduct the simulation using data from ONEDAYLIFE. After removing dialogues with only one utterance, there are a total of 290 dialogues between 25 agents. We regenerate the whole dialogue $D_{ij}^t$ between two agents $A_i$ and $A_j$ at time point $t$. $A_i$ and $A_j$ take turns to generate utterances until one has no reply or the dialogue reaches 16 turns. At each turn, the LLM is provided with the speaker's persona $P_i$, memories $M_i^t$, location, status $S_i^t, S_j^t$ at time $t$, and dialogue histories between the two agents $D_{ij}^k, k \in 1, 2, ..., K^{t-1}$.

We use GPT-3.5-turbo as the backbone LLM throughout the SDR framework, which was used in (Park et al., 2023) (Origin). To compare with a stronger baseline, we reran the simulation from Origin but generated three candidates for each $U_c$ and selected the best one judged by the LLM, denoted as Baseline. More detailed settings are provided in Appendix C.1.

In the simulation framework, each generated dialogue at time $t$ can potentially alter the memory, location, and status of the agents subsequently, thus changing the following dialogues after $t$. To ensure a fair comparison with Origin, we opted not to regenerate new memories, locations, or statuses for agents following their conversations. Instead, we treated each dialogue as a distinct, standalone example.

---

[3] https://platform.openai.com/docs/guides/text-generation/parameter-details

## 3.2 Evaluation

We conduct a corpus-level evaluation on three key aspects: diversity, consistency, and factualness, corresponding to the three error types – repetition, inconsistency, and hallucination – we aim to address.

**Diversity**   We employ widely used metrics Distinct-N (Li et al., 2016) and Semantic Distance (Distance) (Dziri et al., 2019) for diversity evaluation. Please refer to Appendix C.2 and the original paper for more details.

To further analyze the diversity from the perspective of each individual agent, we proposed an agent-based metric, Agent Diversity (Agent Div). This metric is based on the idea that an agent should adjust the conversation content according to the identity of the listener and not always talk about the same thing with everyone. Agent Div is calculated as the average of each agent's Agent Div, which is equal to one minus the normalized similarity between dialogues of the same agent. The algorithm is provided in Appendix C.2.

**Consistency and Factualness**   We assess the corpus-level consistency and factualness by GPT-4 and human evaluation.

**- GPT-4:** Recent works (Liu et al., 2023; Mendonça et al., 2023; Luo et al., 2023; Gao et al., 2023) have shown that GPT-4 evaluation correlates more closely with human judgments. Besides, it has good properties on scalability and reproducibility. Thus, we select GPT-4 for this challenging task. The process to retrieve evidence dialogues is the same as the Screening stage described in Sections 2.1. Then, GPT-4 is utilized to score the consistency and factualness of the current dialogue on a scale from 1 to 10. The error rate represents the ratio of dialogues receiving a score below a threshold of 8, determined based on our empirical observations.

**- Human Evaluation:** We conduct a human evaluation on the most challenging final 10% of the conversations. We recruited two annotators and provided them with the exact same prompt as GPT-4 to rate the consistency and factualness. Annotators were compensated at the minimum hourly wage for 20 hours of work. The average scores are reported in this paper. Following previous works (Ziems et al., 2022; Maronikolakis et al., 2022; Riley et al., 2023), we calculate the intraclass correlation coefficients (ICC) (Shrout & Fleiss, 1979; McGraw & Wong, 1996) to measure the annotator agreement. The ICC for factualness and consistency are 0.47 and 0.44, respectively, representing moderate agreement.

**Fluency**   We utilize the perplexity derived from GPT-2 as the fluency metric. We have not stressed fluency evaluation, as our observations indicate that all generated dialogues are highly fluent and grammatically correct.

## 4   Result and Discussion

| | Diversity | | | Factualness | | Consistency | | Fluency | Turns | Words |
|---|---|---|---|---|---|---|---|---|---|---|
| | Distinct - 1 / 2 / 3 | Distance | Agent Div | Score | Error (↓) | Score | Error (↓) | PPL (↓) | | |
| Origin | 0.117 / 0.473 / 0.726 | 0.234 | 0.454 | 8.58 | 24.5% | 8.17 | 37.2% | 20.37 | 9.6 | 25.4 |
| Baseline | 0.124 / 0.469 / 0.718 | 0.274 | 0.475 | 8.77 | 25.5% | 8.10 | 39.7% | 20.18 | 15.5 | 29.3 |
| SDR (Ours) | **0.132 / 0.521 / 0.773** | **0.311** | **0.502** | **8.89** | **19.0%** | **8.27** | **32.4%** | **19.73** | 10.3 | 42.5 |

Table 1: Corpus-level (multi-dialogues) evaluation. Turns and Words refer to the average number of turns per dialogue and words per turn. The best number is in bold.

**SDR Achieves the Best Diversity, Factualness, Consistency, and Fluency in Multi-dialogue Contexts**   As illustrated in Table 1, our SDR framework (Ours) excels in achieving superior corpus-level dialogue diversity, factuality, consistency, and fluency. SDR maintains an average number of turns similar to Origin, contrasting with the Baseline that tends to extend until reaching a predetermined maximum turn count (16). While SDR does not

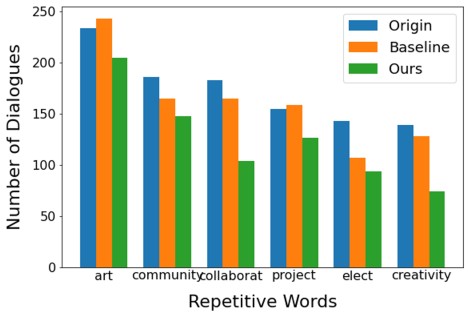

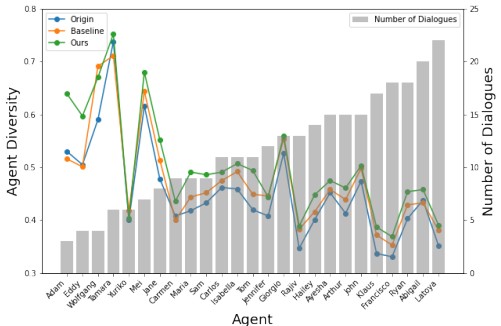

(a) The number of dialogues containing the 6 most repetitive words from TF-IDF.

(b) Comparison of Agent Diversity and the number of dialogues each agent involved.

Figure 5: Diversity Analysis

have a lengthy number of turns, each utterance conveys more comprehensive information, evidenced by a higher word count per turn in SDR.

**SDR Significantly Reduces Keyword Repetition**   Figure 5a demonstrates how our approach effectively reduces the repetition of the most frequently used keywords. To delve deeper into the occurrence of repetition across all dialogues, we conducted an analysis focusing on keyword frequency. Specifically, we determined the keywords by TF-IDF scores and counted the number of dialogues consisting of the top 6 noun keywords[4]. Figure 5a showcases that our method substantially decreases the frequency of dialogues mentioning key terms, particularly for "creativity" and "collaboration." Compared to the Origin, the number of dialogues featuring these keywords has been reduced by up to 47% and 44%, respectively.

**Agent Div Negatively Correlated to the Number of Initiated Dialogues**   We analyze the relation between Agent Diversity and dialogue volume in Fig. 5b. We found that, despite a few exceptions, the Agent Div has a negative correlation with the number of dialogues the agent involved. Although there are a few exceptions, the figure generally shows that the more the agent talks, the lower the diversity they have.

|  | Factualness | | Consistency | |
|---|---|---|---|---|
|  | GPT4 | human | GPT4 | human |
| Origin | 8.34 | 6.90 | 7.41 | 5.00 |
| SDR (Ours) | **8.96** | **8.57** | **8.18** | **6.14** |

Table 2: Human evaluation on the last 10% of dialogues. The best result is in bold.

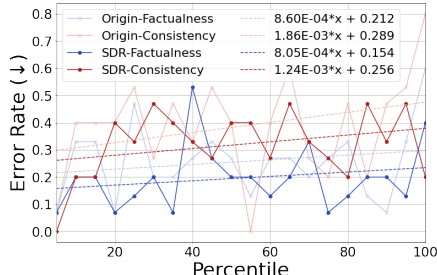

Figure 6: Error rate trends over time.

**SDR Enhances Consistency and Reduce Hallucination**   As illustrated in Table 2, the human evaluation reflects the same trend as the GPT-4 assessment, showing SDR is superior to Original in both factualness and consistency. Additionally, it is observed that the scores for factualness are consistently higher than those for consistency in both the human and

---

[4]Since "collaboration" and "election," have multiple variations, we use the root forms of these words, "collabora" and "elect," to ensure a more accurate representation of their usage across all dialogues.

| Prompt Info | Prompt Type | Diversity | | | Factualness | | Consistency | | Fluency | Turns | Words |
|---|---|---|---|---|---|---|---|---|---|---|---|
| | | Distinct - 1 / 2 / 3 | Distance | Score | Error (↓) | Score | Error (↓) | PPL (↓) | | |
| Origin | | **0.445** / 0.724 / 0.886 | 0.212 | 8.34 | 31.0% | 7.41 | 55.2% | 22.2 | 8.1 | 24.4 |
| Baseline | | 0.323 / 0.709 / 0.869 | 0.238 | 8.07 | 44.8% | 7.72 | 41.4% | 20.1 | 15.0 | 31.2 |
| All | Task | 0.278 / 0.742 / 0.918 | 0.306 | 8.45 | 31.0% | 8.21 | 34.5% | 21.3 | 10.6 | 36.0 |
| All | Persona | 0.286 / 0.751 / 0.917 | 0.288 | 8.52 | 27.6% | 7.79 | 41.4% | 20.3 | 9.9 | 43.9 |
| All | Mixed | 0.292 / 0.744 / 0.919 | 0.303 | 8.66 | 27.6% | 8.21 | 44.8% | **19.0** | 11.0 | 41.8 |
| - background | Mixed | 0.303 / 0.751 / 0.921 | **0.338** | 8.79 | 32.1% | 8.14 | 32.1% | 20.4 | 9.4 | 33.7 |
| - memory | Mixed | 0.349 / **0.778** / **0.931** | 0.305 | **8.96** | **17.9%** | 8.18 | 35.7% | 19.2 | 10.6 | 44.1 |
| - history | Mixed | 0.319 / 0.774 / 0.926 | 0.292 | 8.38 | 31.0% | **8.69** | **31.0%** | 20.2 | 9.9 | 42.8 |
| - status | Mixed | 0.271 / 0.717 / 0.898 | 0.257 | 8.39 | 25.0% | 8.21 | 35.7% | 19.6 | 9.9 | 49.0 |

Table 3: Ablation study on the last 10% of conversations in ONEDAYLIFE. The best number is in bold, and the second best is underlined.

**GPT-4 evaluations.** Notably, the GPT-4 scores are higher than human evaluations in all cases, which might indicate that there are potential errors not detected by GPT-4.

Furthermore, we plot the error rate changes across the percentile of the number of dialogues, as shown in Fig. 6. We observed that the errors gradually increase toward the higher percentile, especially in the last one, suggesting that inconsistencies or contradictions become more frequent at higher percentiles. The analysis of error rate trends shows that SDR generally exhibits lower error rates than Origin in both factualness and consistency. Additionally, the error rate slope of Consistency for Origin is 1.5 times that of SDR. This disparity indicates a more pronounced error propagation in Origin, highlighting the effectiveness of our method in maintaining dialogue integrity over time.

**SDR can Balance Diversity and Faithfulness** Table 3 shows the ablation study for various prompt designs. The ablation study is conducted on the last percentile of conversations, where the origin got the worst factualness and consistency scores. We first identify the benefit of using diverse prompt types. Randomly picking from a structured task-oriented prompt or a persona-based narrative prompt yields better or comparable results than using either of them. We also investigate whether all information in the original prompt is necessary. Surprisingly, we found that prompts excluding memory often outperformed others in most aspects. This outcome seems counterintuitive, as memory is generally considered crucial for preventing hallucinations and ensuring consistency. However, memory can impose a strong constraint that may reduce conversational diversity. By opting for a no-memory prompt, we open the door to more diverse conversational content. Our SDR framework ensures that the utterance $U_c$ can be consistent with previous dialogues and free from critical factual errors. This approach allows us to strike a balance between diversity and faithfulness in multi-agent multi-dialogue generation.

**Effectiveness of Screening and Diagnosis Stages** We evaluate whether our SDR framework can successfully detect potential problems by calculating the precision (P), recall (R), and F1 score for the output from the Screening and Diagnosis stages, using a subset of ONEDAYLIFE. We particularly emphasize on recall, as our primary objective is to identify potential issues at these stages. We manually annotated 50 utterances as either problematic or non-problematic and processed them through our framework up to the Diagnosis stage. The results validate the effectiveness of the procedures in the Screening and Diagnosis stages, with P, R, and F1 scores of 64.7%, 88.0%, and 74.6%, respectively. We also show an example that fails to pass all three pipelines in Table 5 in Appendix D.1.

**NLI-G Study** We assess the effectiveness of NLI-G using evaluation scores on Origin. We compare the factualness and consistency scores based on two sources of evidence: (1) evidence retrieved by NLI-G and (2) evidence from the agent's previous $k$ dialogues (denoted as *Prev*). In both cases, the number of evidence is five. In Fig. 7, the left figure shows that the scores using NLI-G evidence are generally lower than those with *Prev* evidence (-0.82 and -0.62 for factualness and consistency, respectively). The lower score indicates that more effective evidence is retrieved, where "effective evidence" refers to dialogues that the candidate utterance $U_c$ contradicts. Therefore, it indicates NLI-G's

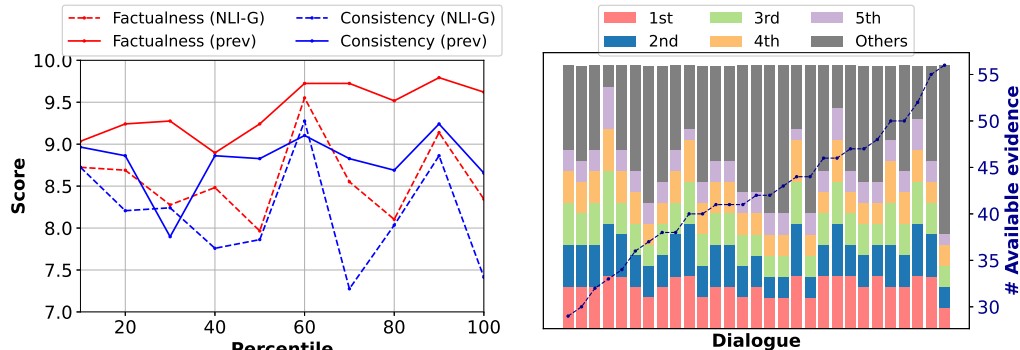

Figure 7: The study of NLI-G's effectiveness and robustness. **Left:** Scores using evidence retrieved by NLI-G or previous dialogues (prev). Each data point is the mean score of each percentile. **Right:** Frequency distribution of dialogues retrieved by NLI-G. Each bar represents a dialogue in the last 10% of ONEDAYLIFE, sorted by the number of available evidence dialogues.

superior evidence retrieval ability. Additionally, we examine NLI-G's retrieval variance by evaluating the last 10% of dialogues five times, counting the frequency of evidence dialogues. The right figure illustrates the composition of each dialogue's evidence count, with colored sections representing the proportions of the top five pieces of evidence (average colored area: 61.8% of the bar) and gray for the others. This demonstrates NLI-G's consistency in retrieving similar dialogues across different trials, even when the number of available evidence dialogues – equal to the quantity of past dialogues involving either of the agents – exceeds 50 in later dialogues.

## 5 Related Work

**LLM-powered Agents**    A typical LLM-powered agent encompasses predefined or dynamically generated prompt templates to leverage the LLM's capabilities and achieve specific functionalities, such as making decisions or interacting with its surroundings (Zhao et al., 2023). Wang et al. (2023a) can autonomously navigate the game world, maintaining a skill library to expand its problem-solving scope. Richards & et al. (2023); Nakajima (2023) employ chain-of-thought (CoT) (Wei et al., 2022) to provide automated solutions for designated tasks. Nevertheless, a singular agent undeniably has its limitations. The collective intelligence formed by multiple agents can yield results greater than the sum of its parts. Hong et al. (2023) integrates specialized human SOP expertise, successfully accomplishing intricate software development. Chen et al. (2023b) focuses on communication mechanisms among heterogeneous agents to enhance decision-making efficacy. Differing from all previous works, we solve problems of the conversational content in multi-agent dialogues.

**Revision Strategy**    Revising undesired LLM outputs is crucial for enhancing generation quality. Revision strategies span from simple resampling (Wang et al., 2023b) to leveraging feedback from self-generated signals or results from external modules (Schick et al., 2023), or a mixture of them. The type of revision feedback could be scalar values (Shinn et al., 2023) or natural language (He et al., 2023).By incorporate score threshold with explanation, we provide quality control for generated comments. Most revision strategies revise LLM outputs for a definite task to improve task success rates (Madaan et al., 2023; Skreta et al., 2023; Tyen et al., 2023). On the contrary, we aim at revising open-domain dialogue utterances on the fly. The revision strategies for generation tasks directly take full simulation trajectories (all past generated records) as input (Shinn et al., 2023; Saunders et al., 2022). However, LLMs struggle to properly process the ever growing history, ignoring the subtle information in the middle of lengthy inputs (Liu et al., 2024). We introduce a screening stage to narrow the scope of simulation history, enabling a more efficient revision strategy for scaled simulations.

To be more specific, we incorporate pretrained modules and LLM's reasoning ability without training a series of cumbersome classifiers (Bakhtin et al., 2022). In addition, many previous works focus on only one revision direction, potentially guided by the end goal of the goal-oriented task. Li et al. (2023) explicitly tracks goal-related states. Nevertheless, SDR adheres to the divide-and-conquer philosophy, breaking down a problem into manageable parts for more effective resolution.

**Machine-Generated Dialogues**   Our work is distinct from other machine-generated multi-session dialogues in several aspects. We first emphasize that the utterances in ONEDAYLIFE are generated iteratively by each agent, rather than through full conversation syntheses like in SODA (Kim et al., 2023), PLACES (Chen et al., 2023a), and Dialog Inpainting (Dai et al., 2022). This approach more realistically simulates agent behaviors but presents challenges in maintaining global coherence (Zhou et al., 2024). Besides, the conversations in ONEDAYLIFE exhibit causal relationships, unlike the independent dialogue pieces in other datasets. These specific properties make ONEDAYLIFE particularly suitable for our study on the authenticity of LLM agents, and in uncovering issues inherent in multi-agent, multi-session dialogues.

## 6    Limitations

The primary limitation of our research lies in the considerable cost associated with employing GPT-3.5-turbo as the backbone language model, which might not be feasible for all research budgets. Additionally, our findings are confined to ONEDAYLIFE. Any inherent flaws in the original data are thus carried over into our simulated conversations, potentially affecting evaluation scores. Moreover, as ONEDAYLIFE lacks annotations, our evaluation metrics – spanning automatic, LLM-based, and human metrics – may not fully capture all possible aspects of the dialogues.

Our work also has a few limitations. Firstly, we simulate individual conversations rather than full-day interactions, which significantly reduces costs but also limits our ability to replicate the comprehensive dynamics reported in the original study. Secondly, although there is a chance that an error-free utterance might be sent to the regeneration stage, we do not consider it a severe problem because regeneration does not inherently cause errors, and the regenerated utterance will still undergo the entire SDR pipeline. Lastly, our SDR framework is specifically designed for open-domain dialogues. Its applicability to goal-oriented dialogues remains unexplored, which we leave for future work.

## 7    Conclusion

In conclusion, our study introduces and validates a novel multi-agent dialogue simulation framework equipped with a Screening, Diagnosis, and Regeneration (SDR) mechanism, addressing the inherent challenges in multi-agent, multi-session dialogues such as repetition, inconsistency, and hallucination. By meticulously analyzing the dialogues generated in the ONEDAYLIFE simulation, we identified significant issues that compound over time, negatively impacting dialogue quality. Our proposed SDR framework effectively reduces these problems and enhances dialogue diversity, consistency, and factualness, as evidenced by automatic metrics, GPT-4 assessments, and human evaluations. We successfully reduce keyword repetition, maintain dialogue integrity over multiple sessions, and balance conversational diversity with the faithfulness of multi-agent dialogue simulation. Our work not only highlights the previously unaddressed challenges in multi-agent dialogue simulation but also sets a new standard for dialogue quality, paving the way for more advanced and realistic simulations in future research.

## Acknowledgement

This work was supported by JSPS/MEXT KAKENHI Grant Numbers JP23K28139 and JP22H05015, and the commissioned research (No. 225) by the National Institute of Informa-

tion and Communications Technology (NICT), and the Institute of AI and Beyond of the University of Tokyo.

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

## A Human-Like Multi-Agents

### A.1 From General LLM to Individual Persona

Generative Agents (Park et al., 2023) introduces a two-component architecture for creating personalized, dynamic human-like agents: a string-based memory base and an LLM-driven cognitive function set. The memory base stores memories over time, aiding in the development of diverse agents, while the LLM-centric cognitive functions simulate human capabilities like reflection, planning, and reaction. Combining these, the LLM uses memory-derived context to tailor knowledge extraction and response formulation, ensuring agent-specific behaviors.

In a scenario where two agents initiate a dialogue, each iteratively produces utterances informed by context like location, status, and memories. The dialogue function uses a specific prompts like: "Based on the [...] information, what will [name] say next?" For more details on Generative Agents and the simulation of 25 agents in a village, please see the original paper.

### A.2 Conversations and Transmission

The memory capabilities enable the transmission of information to **both** agents involved in a dialogue section. However, this also means that undesirable dialogue content could spread in the same manner.

Figure 2 illustrates this using keyword spreading as an example. The bar chart shows the proportion of dialogues containing the keyword (in red) compared to the total number of conversations (in blue), highlighting a swift escalation, sometimes reaching 100%. Chord diagrams further reveal that initially, only a few agents act as propagators, but as the day progresses, the majority become involved in similar actions, as indicated by the variety of line colors. Consequently, the dialogue topics become repetitive and less believable. This example underscores the scale of the spreading and the associated risks, emphasizing the need for a dynamic correction approach.

## B SDR Details

### B.1 Dynamic Similarity

We employ dynamic similarity threshold to determined the repetition of utterances. The value of $\theta_{sim}$ is adjusted based on specific criteria: it is decreased if the similar utterance originates from the same agent $A$ as $U_c$ and is in the current dialogue $D_t$, which indicating repetition. Conversely, a higher threshold is more acceptable if the similar utterance is from $A$ but a different dialogue. This process is formalized as Eq. 1.

$$\theta_{sim} = \begin{cases} \theta + \alpha & \text{if same } A, \text{ different } D \\ \theta - \alpha & \text{if same } A, \text{ same } D \\ \theta & \text{otherwise,} \end{cases} \tag{1}$$

### B.2 The Extracted Graphs in NLI-G

We demonstrate an example of the personal information graph extracted from a dialogue. Table 4 lists a pair of dialogue and the extracted graph.

| | |
|---|---|
| **Raw Dialogue** | **Giorgio Rossi**: Excuse me, are you John Lin?
**John Lin**: Yes, that's me. How may I help you?
**Giorgio Rossi**: I noticed that you're very knowledgeable about medications and treatments. I've been curious about some of the latest developments in that field. Would you mind sharing some of your insights with me?
**John Lin**: Of course, I'm happy to share what I know. What specifically are you interested in?
**Giorgio Rossi**: Well, I'm particularly intrigued by how some medications are being developed based on mathematical patterns found in nature. Do you have any knowledge about that?
**John Lin**: Hmm, that's not really my area of expertise. But I do know that Jennifer Moore, the artist who's running for local mayor, is also a scientist who incorporates mathematical patterns into her work. Maybe she would have more information on that. |
| **Extracted Graph** | [
    ["Giorgio Rossi", "is interested in", "medications and treatments"],
    ["John Lin", "is", "John Lin"],
    ["Giorgio Rossi", "discusses interests with", "John Lin"],
    ["John Lin", "is knowledgeable about", "medications and treatments"],
    ["Giorgio Rossi", "is interested in", "mathematical patterns found in nature"],
    ["John Lin", "does not have expertise in", "medications developed based on mathematical patterns found in nature"],
    ["Jennifer Moore", "is mentioned by", "John Lin"],
    ["Jennifer Moore", "has role", "artist"],
    ["Jennifer Moore", "is running for", "local mayor"],
    ["Jennifer Moore", "has profession", "scientist"],
    ["Jennifer Moore", "incorporates", "mathematical patterns into work"],
] |

Table 4: Example of a graph extracted from raw dialogue in the NLI-G module.

## C   Experiment Details

### C.1   Hyper-Parameters

At the Screening phase, we set $K_{sim}$ to 5 and threshold $\theta$ to 0.85, $\alpha$ to 0.05, $\theta_{force}$ to 0.95 for repetition detection. Utterances shorter than 10 words are excluded from the repetition screening, thus likely preserving the natural "social glue turns". For inconsistency detection, we adapt the DeBERTa-based NLI model, pretrained on multiple NLI datasets Laurer et al. (2023). The $\theta_{nlig}$ is as high as 0.98 as there are a lot of false positive, and we select top $K_{nlig} = 3$ dialogues as the potential contradictory dialogue evidences. The threshold for harmful hallucination likelihood $\theta_{fact}$ is set to 6. The number of diagnose trials $N_{diag}$ is 3, and we selected the LLM feedback with the highest score. If there are more than one feedback that have the same highest score, we chose the longer one. The regeneration threshold $\theta_{regen}$ is 8. Our SDR procedure will terminated if no comments are found or until reaching $R = 2$ rounds. We use GPT-3.5-turbo-0613 for running the baseline and SDR. Note that ONEDAYLIFEwas generated by GPT-3.5-turbo before April, 2023.

For GPT-4 evaluation, $\theta_{nlig}$ is set to 0.99 and $K_{nlig}$ is 5. We use the model GPT-4-0613.

### C.2   Evaluation

**Distinct-N (Li et al., 2016)**   Distinct-N calculates the ratio of unique N-grams in a given text. However, it may not fully capture the corpus-level dialogue diversity, particularly when each dialogue has longer utterances, since individual dialogues typically revolve around a single topic. To address this, we apply Distinct-N to summaries of dialogues, generated by a pretrained dialogue summarization model. This approach allows us to more

effectively gauge the thematic diversity of dialogues at the corpus level across multiple conversations.

**Semantic Distance (Dziri et al., 2019)** To complement the word-based Distinct-N metric, we measure the Semantic Distance on embedding space. Specifically, we calculate the cosine similarity between dialogue embeddings. Semantic Distance is then determined as $1 - similarity$.

**Agent Diversity** The algorithm of Agent Diversity is shown in Algorithm 1. $Emb(\cdot)$ is a speaker dialogue embedding calculated from length-weighted utterance embeddings in the dialogue.

---

**Algorithm 1:** Agent Diversity (Agent Div)

---

**Data** : Agents $A_i$ for $i \in \{1, 2, ..., N\}$,
          Dialogues $D_{ij}^k$ for $j \neq i$ and $0 < k \leq K_{ij}$

1  // Calculate $AgentDiv_i$ for each agent $A_i$;
2  **targets** $\leftarrow \{j \mid K_{ij} \neq 0\}$;
3  sims $\leftarrow 0$;     pairs $\leftarrow 0$;
4  **for** *each unique pair (p, q) in* **targets** **do**
5      $E_p \leftarrow \{\text{Emb}(D_{ip}^k) \mid k \in \{1, 2, \ldots, K_{ip}\}\}$;
6      $E_q \leftarrow \{\text{Emb}(D_{iq}^k) \mid k \in \{1, 2, \ldots, K_{iq}\}\}$;
7      $s_{pq} \leftarrow \frac{1}{K_{ip} K_{iq}} \sum_{a=1}^{K_{ip}} \sum_{b=1}^{K_{iq}} \text{CosSim}(E_{p_a}, E_{q_b})$;
8      sims $\leftarrow$ sims $+ s_{pq}$;     pairs $\leftarrow$ pairs $+ 1$;
9  **end**
10  $AgentDiv_i \leftarrow 1 - \frac{sims}{pairs}$
11  // Averaging $AgentDiv_i$ for all agents
12  $AgentDiv \leftarrow \frac{1}{N} \sum_{i=1}^{N} AgentDiv_i$ ;

---

## D   Additional Analysis and Examples

### D.1   Output of Diagnosis Stage

We show the output of Diagnosis stage in Table 5.

### D.2   GPT4 Score and Dialogue Length

We investigate if there are biases between the dialogue length and the score given by GPT-4. We use Pearson Correlation Coefficient (Rodgers & Nicewander, 1988) and the Spearman Rank-Order Correlation Coefficient (Spearman, 1961) to examine the correlation between scores (factualness and consistency) and the dialogue statistics (number of utterances and words). The results are shown in Table 6, and it shows no or low correlations between them.

### D.3   Problematic Dialogues in ONEDAYLIFE

Below are four dialogue snippets from ONEDAYLIFE involving the four keywords "math", "poetry", "artwork", and "collaborat".

One unnatural point is that **agents from diverse backgrounds all show interest in collaborating on poetry and artwork (and mathematical patterns)**: Carmen (a shopkeeper) [D1,D2,D3], Latoya (a photographer) [D1,D2,D4], Hailey (a writer) [D2], Arthur (a bartender) [D2], Tamara (a children's book author) [D3], Giorgio Rossi (a mathematician) [D3], Abigail Chen (a digital artist and animator) [D3], Francisco Lopez (an actor and comedian) [D3,D4], Ryan Park (a software engineer) [D4], Rajiv (a painter) [D4].

| | | |
|---|---|---|
| Context | | Time: 2023-02-13 11:55:40 |
| | | Location: store in The Willows Market and Pharmacy |
| | | Status:
John: performing special tasks and handling customer queries (checking the inventory)
Yuriko: taking a break and having lunch (researching the local mayor election) |
| | | Dialogue Context:
John: Hey there, Yuriko. Looks like we're both keeping busy with work and classes.
Yuriko: Yes, it seems like we're both trying to stay on top of things. I'm actually taking classes on new tax laws.
John: That's interesting. I'm taking classes on new medications and treatments for my pharmacy.
Yuriko: I've heard about some exciting new research on tax incentives for small businesses. Have you heard anything interesting lately?
John: Actually, I've been asking around about who's running for mayor next month. Have you heard anything?
Yuriko: Yes, I've heard that Sam Moore is running. He has a vision for investing in the local economy and fostering creativity and innovation. What do you think about that? |
| $U_c$ | | John: I actually have a friend who's running for mayor, Jennifer Moore. She's an artist and is interested in developing medications based on mathematical patterns found in nature. |
| | Score | Reason |
| Repetition | 8 | The response contains some repetitive statements that are already known from the previous conversation. For example, John Lin already mentioned that he has a friend named Jennifer Moore who is running for mayor and is interested in medication development based on mathematical patterns. Restating this information in the response is redundant. Additionally, John asking for more information from Jennifer Moore is already implied by him stating that she is his friend. These repetitive statements make the response less natural. |
| Consistency | 8 | John Lin's candidate response refers to his friend Jennifer Moore running for mayor, while in the past dialogue, he mentioned Sam Moore as a candidate. |
| Hallucination | false | Jennifer Moore mentioned that she is not interested in politics and prefers to focus on her art and her relationships with the people she cares about, indicating that she would not agree with the statement that John Lin's friend is running for mayor. |

Table 5: Diagnosis stage output.

|  | Pearson Coef. | | Spearman Coef. | |
|---|---|---|---|---|
|  | # Turn | # Word | # Turn | # Word |
| Factualness | 0.054 | 0.086 | 0.026 | 0.039 |
| Consistency | -0.025 | -0.021 | -0.024 | -0.014 |

Table 6: The correlation coefficients between GPT-4 scores and dialogue statistics on Origin.

Another unnatural point is the **variation in content and participants involved in the collaborations**. For example, people collaborate on poetry and artwork in each dialogue are:

[D1] Carmen, Latoya

[D2] Carmen, Latoya | Hailey, Arthur

[D3] Carmen, Tamara, Giorgio, Abigail, Francisco

[D4] Latoya, Ryan | Rajiv, Francisco

Furthermore, while Carmen does not have other conversations between D1 and D2, the collaborations she mentions differ in the two dialogues.

[D1]

1. Rajiv - art projects and medicinal development
2. Latoya - poetry and artwork | mobile app
3. Arthur - mixology

[D2]

1. Tamara and Rajiv - creating job opportunities and expanding book sales online
2. Latoya - poetry and artwork
3. Giorgio and Arthur - mixology and medicinal development

Dialogue Snippets:

[D1] Time: 2023-02-13 14:43:20

Arthur (a bartender): Excuse me, Carmen. I couldn't help but overhear your conversation about potential **collaborat**ions. ... Additionally, I'm passionate about creating job opportunities for young people in the community and would love to brainstorm with you on how we could work together on that. What do you think?

Carmen (a shopkeeper): That sounds great, Arthur! I'm actually meeting Rajiv Patel (a painter) here later to discuss potential **collaborat**ions on art projects and medicinal development using **math**ematical patterns found in nature. ... I'm also discussing potential **collaborat**ions with various individuals about promoting art in low-income communities and supporting local businesses. And we have plans for a **poetry** and **artwork** project with Latoya Williams (a photographer). Additionally, we're discussing potential **collaborat**ions on creative projects such as a mobile app for local businesses and community events. I would love to hear more about your mixology ideas and how we might be able to **collaborat**e on that as well.

[D2] Time: 2023-02-13 14:57:20

Hailey (a writer): I was actually just talking to Arthur Burton (a bartender) about potential **collaborat**ions in mixology, art, **poetry**, improv, and medication development based on **math**ematical patterns in nature. I invited him to be a guest on my podcast to contribute unique cocktail recipes. And then I saw you and thought it would be great to catch up.

Carmen (a shopkeeper): Wow, that sounds really interesting! I'm actually discussing potential **collaborat**ions with Tamara Taylor (a children's book author) and Rajiv Patel (a painter)

on creating job opportunities for young people in the community, as well as expanding book sales online. I'm also working with Latoya Williams (a photographer) on a **poetry** and **artwork** project, and considering **collaborat**ions with Giorgio Rossi (a mathematician) and Arthur Burton (a bartender) on mixology ideas and medicinal development using **math**ematical patterns found in nature. We all plan to grab a drink here after Rajiv's first solo show.

[D3] Time: 2023-02-13 15:05:20

Jennifer (a painter): That sounds like a great idea! I'd love to **collaborat**e with you and other local artists. Do you have any other projects or **collaborat**ions in mind?

Tamara (a children's book author): Well, I'm considering **collaborat**ions with Carmen Ortiz (a shopkeeper), Giorgio Rossi (a mathematician), Abigail Chen (a digital artist and animator), and Francisco Lopez (an actor and comedian) for projects involving **poetry**, **artwork**, and potentially other fields like science and **math**. I'm also interested in attending political discussions and expanding my book sales online. What about you?

[D4] Time: 2023-02-13 18:16:20

Latoya (a photographer): That sounds like a great idea. And I'm also open to **collaborat**ing with you on exploring connections between **math**, nature, and art. I'm actually discussing a similar project with Ryan Park (a software engineer).

Rajiv (a painter): That's great to hear. And if you're interested, Francisco Lopez (an actor and comedian) and I are also exploring a project involving **poetry**, **artwork**, and **math**ematical patterns. So there's definitely a lot of potential for **collaborat**ion and creativity in this space.

## E   Prompt Examples

### E.1   Repetition Check Prompt

Context for the task:
[Speaker's background]

Here are some conversation histories between various people:
[Speaker]:**
Time: $t_0$
[Evidence Dialogue $D_{t_0}$ ]
...
Time: $t_k$
[Evidence Dialogue $D_{t_k}$ ]

[Speaker] is about to say the following sentence ('the response') next in the latest session:
[ $U_c$ ]

---
# Task: Please identify any "unnatural points" in 'the response'.
An "unnatural point" refers to redundancies or repetitive statements made in 'the response' when considering the context of the previous conversations.
On a scale of 1 (no unnatural point) to 10 (the most significant of unnatural point), rate the likely significant score of 'the response'. And explain the reason for the score.

Output format: Output a json of the following format:
{
    "reason": "point out the unnatural point and your reason for the score",
    "score": "<json integer>"
}

### E.2   Consistency Check Prompt

# Context
**Background:**
[Speaker's background]

**Past Dialogues involving [Speaker]:**
Time: $t_0$
[Evidence Dialogue $D_{t_0}$ ]
...
Time: $t_k$
[Evidence Dialogue $D_{t_k}$ ]

**Current Dialogue between [Speaker] and [Listener]:**
Time: $t$
[Current Dialogue $D_t$ ]

**Candidate Response:**
[Speaker] is planning to say: [ $U_c$ ]

---
# Task
Determine if there is any contradiction between the candidate response and the past dialogue/character background. Do not consider the absence of a repeated mention as an inconsistency. Ignore statements that are situational or not meant to be taken literally. Let's think step by step.

**Output a JSON object:**
{
    "Contradiction?": <true/false>,
    "Details": "<Specify any contradictions, if any>"
}

## Example
[An example including Past Dialogues, Current Dialogue, Candidate Response, and Output]

### E.3   Agent Agreement Prompt

Context for the task:

Here is a brief description of [Mentioned Agent].
[Background of the Mentioned Agent]

Here is the memory that is in [Mentioned Agent]'s head:
[Memory of the Mentioned Agent]

Here is the previous conversation between [Mentioned Agent] and [Speaker]:
Time: $t_{-1}$
[Evidence Dialogue $D_{t_{-1}}$ ]

[Speaker] just mentioned the following statement about [Mentioned Agent]:
[ $U_c$ ]

---
# Task: Based on the information provided above, would [Mentioned Agent] agree with the statement?

Output format: Output a json of the following format:
{
    "agreed": "<json Boolean>",
    "reason": "the reason that led [Mentioned Agent] to make the judgment"
}

## E.4 Revise Utterance Prompts

### E.4.1 Persona-based Narrative

Your name is [Speaker].
Your background is as follows:
[Background of the Speaker]

You are engaged in a conversation with [Listener], and here is the content of the dialogue so far:
[Current Dialogue $D_t$ ]

# Task:
Consider whether you would respond to [Listener]. If you choose to reply, what would you say? Would your response aim to conclude the conversation?
You might consider saying "$U_c$", but it has some issues, for instance:
[Reason]
Here are some suggestions for your reference:
[Suggestion]
If the response is redundant or repetitive, you can end the current dialogue.

**Output a JSON object:**
{
    "Response": "<your reply as [Speaker] (if any)>",
    "The conversation ends with [Speaker]'s utterance": <true/false>
}

### E.4.2 Structured Task-oriented

# Contextual Information:
**Introduction:**
[Background of the Speaker]

**Current Dialogue between [Speaker] and [Listener]:**
[Current Dialogue $D_t$ ]

# Task:
Assuming the role of [Speaker], consider whether you would respond to [Listener]. If you choose to reply, what would you say? Would your response aim to conclude the conversation?
You might be considering saying something that has some issues, such as:
[Reason]
Here are some suggestions for your reference:
[Suggestion]
If the response is redundant or repetitive, you can end the current dialogue.

**Output a JSON object:**
{
       "Response": "<your reply as [Speaker] (if any)>",
       "The conversation ends with [Speaker]'s utterance": <true/false>
}

