# OpenReview forum: "Cohesive Conversations: Enhancing Authenticity in Multi-Agent Simulated Dialogues"
_colmweb.org/COLM/2024/Conference — COLM_

### Official Review · Reviewer_8Cvc · 2024-04-23

**Rating:** 8
**Confidence:** 3
**Ethics Flag:** 1

**Summary:**

The authors analyze the data generated by the simulation framework published by Park et al., and propose a heuristic strategy to detect and correct errors/problematic behavior in the dialogues.  Here, the focus is not on the quality of a short-term dialogue but on problems which only become visible when multiple sessions are investigated. The authors demonstrate improvements by some objective quality measures for the corrected dialogues. The paper is clearly written, the work is original as such day-long interactions between LLM-based agents are a new thing. I'd consider the work as significant because I am not aware of other studies that analyze and improve the quality of the simulated dialogues and this is clearly a next step towards simulating human behavior more realistically.

**Reasons To Accept:**

I suggest that the paper should be accepted for the reasons given above in my summary.

**Reasons To Reject:**

I don't see a reason to reject.

---

> ### Author Rebuttal · Authors · 2024-05-31
>
> Thank you for your positive feedback. We're glad you found our work significant, original, and well-written. If you have any further suggestions, we'd be happy to consider them. Thank you again for your review.

---

> > ### Comment · Reviewer_8Cvc · 2024-06-04
> >
> > thank you for your response.

---

### Official Review · Reviewer_Z7Mk · 2024-05-08

**Rating:** 7
**Confidence:** 3
**Ethics Flag:** 1

**Summary:**

The paper proposes a framework to reduce repetition, inconsistencies, and hallucinations in multi-agent simulated dialogues. The framework consists of three steps: screen, diagnose and re-generate. In the screening phase, repetitive utterances are identified by querying a database for similarity with past utterances, inconsistencies are identified using an NLI model, and hallucinations are identified using an LLM prompt on utterances with agent names. In the diagnosis phase, LLM prompts are used to score the flagged utterances. In the re-generation phase, the flagged utterances with scores above a threshold are re-generated using LLM prompts.

This framework is evaluated on the OneDayLife corpus. Repetition is measured with diversity metrics Distinct-N, Semantic diversity and Agent diversity. Inconsistencies and hallucinations are measured with gpt-4 prompts, perplexity and a subset with human evaluations. Results are compared to the baselines of the original dialogue and a re-run of the dialogue where 3 candidates are generated for each utterance and the best one is selected with an LLM prompt. The results show that applying the SDR framework yields better diversity, factuality, consistency and fluency along both automated and human metrics. Ablation studies show that omitting memory from the prompt yields better dialogues across most metrics. A manual binary annotation of a subset of utterances demonstrates the effectiveness of the screening + diagnosis stages. Finally, the experiments on the NLI-G module demonstrate its superiority against a simple baseline of retrieving the 5 most recent utterances and its consistency across multiple trials.

**Questions To Authors:**

# Questions
Q1: How are thresholds calculated?

Q2: In the ablation study, what does memory refer to? How is it different from history and background? What is status?

Q3: For the section "Effectiveness of Screening and Diagnosis Stages", how was this annotation carried out? Were any of these doubly annotated? What does problematic mean?

Q4: For the NLI-G study, how do lower scores demonstrate its effectiveness? What if some of the utterances that were examined actually were factual and consistent, then wouldn't a lower score mean NLI-G is worse?

Q5: What percent of responses to prompts are invalid (i.e., do not conform to the required output fields)?

# Suggestions
S1: Rename title. "Authenticity" is ill defined and is not the focus of the paper.

S2: Table 1: write in the caption what the bolding means.

S3: Table 3: write in the caption what the underlining and bolding means.
:

# Typos:

Even Jennifer has told John that she is not > Despite Jennifer telling John that she is not

whether the mentioned agent agree with > whether the mentioned agent agrees with

The former prompt ask the > The former prompt asks the

SDR Enhance Consistency and Reduce Hallucination > SDR Enhances Consistency and Reduces Hallucination

various prompt design > various prompt designs

Ablation study on the last 10% conversations > Ablation study on the last 10% of conversations

Randomly pick from structured task-oriented prompt or persona-based narrative prompt  > Randomly picking from the structured task-oriented prompt or the persona-based narrative prompt


# Prior work:

- Theory of Mind for Multi-Agent Collaboration via Large Language Models. Li et al, 2023.

- LEGO: A Multi-agent Collaborative Framework with Role-playing and Iterative Feedback for Causality Explanation Generation. He et al, 2023

**Reasons To Accept:**

The strengths of this paper are:
	- The paper motivates the need for the framework with a clear and methodical presentation of the issues in the previous dialogues.
	- The paper includes human evaluations to supplement the automatic metrics, which addresses the concern of circularity in using GPT to both generate and evaluate.
	- The paper presents ablation studies to further understand the different modules of the framework.

**Reasons To Reject:**

The weaknesses of this paper are:
	- The paper evaluates only one corpus. The generalization capability of this framework is unclear.
	- The framework relies on several thresholds across all the stages, but it's unclear how these thresholds are calculated in order to understand how much data is required for "training" the framework on a given corpus.
	- Details are lacking on the human evaluation for a reader to understand the reliability of these annotations. The paper would benefit from reporting the inter-annotator agreement, in addition to briefly reporting how the annotators were trained (if at all). Separately, the paper should indicate the compensation received by the annotators.
	- The conclusion that NLI-G outperforms the "Prev" baseline for evidence selection may be erroneous or would benefit from inclusion of further details. If an utterance actually is factual and consistent, how would a drop in its score demonstrate that NLI-G is better?
	- The paper would benefit from a more comprehensive discussion of prior work.

---

> ### Author Rebuttal · Authors · 2024-05-31
>
> Thank you for the review and constructive suggestions. Below, we address the main concerns:
> ## Generalization Capability
> OneDayLife is so far the only publicly available corpus suitable for evaluating our system, as it provides extensive, multi-session interactions between agents.
>
> Our framework can be effectively applied to other social interaction simulations for several reasons: a) Generic Agent Design: The modules used in our system are common in LLM agent studies, making integration into future works easier. b) Inevitable Errors by LLM Agents: Advanced LLMs may still produce inaccurate content, so the ability to correct such errors in agent interactions is crucial. The three error types we address are common across various social interactions.
> ## Thresholds
> Thresholds were set based on empirical observations. We believe these thresholds should be adjustable according to the simulation's objectives rather than being learned exclusively from a specific corpus.
>
> ## Human Evaluation
> We reported the intraclass correlation coefficients to show the consistency of human evaluation. Annotators were not specifically trained; instead, they received the same instructions and data setup as provided to GPT-4, with procedures for shuffling and anonymizing the content. Annotators were compensated at the minimum hourly wage for 20 hours of work.
> ## NLI-G Explanation
> If the current dialogue is consistent, it should receive a high score, showing no significant difference between NLI-G and Prev. Conversely, if the current dialogue contradicts histories, the lower the score, the more evidence is provided, indicating a better evidence retrieval ability.
> In Fig. 7, each data point represents the mean score for each percentile. While both error-free and erroneous dialogues exist, the average score differences between NLI-G and Prev mainly stem from erroneous examples. Thus, a lower average score suggests NLI-G effectively identifies discrepancies in the dialogue.
> ## Prior Work
> We will append the related work section.
> ## Answers
> Q2. Memory, history, background, and status constitute the context for utterance generation, originating from GenerativeAgents.
>
> Q3. The annotation was done by the author; We manually selected 25 problematic utterances (containing any repetition, inconsistency, or factual error) and 25 issue-free ones. Each has a single binary label.
>
> Q5. We enforce the model to output in a JSON-like format and retry if invalid. Therefore, 100% of them are valid.

---

> > ### Comment · Reviewer_Z7Mk · 2024-05-31
> > **Reviewed author rebuttal**
> >
> > I have read the auth response and leave my scores the same.
> >
> > For generalizability, there are other corpora of agents in goal-oriented chats but it seems this methodology is only applicable to social chats. This limitation should be explicitly stated and explained in the paper. For thresholds, it's unclear whether they result in overfitting to the dataset. For human evaluation, it's unclear why inter-annotator agreement isn't calculated. For NLI-G study, the paper lacks evidence to claim that "the average score differences between NLI-G and Prev mainly stem from erroneous examples".

---

> > > ### Author Response · Authors · 2024-06-01
> > > **Clarification to reviewer's response**
> > >
> > > Thank you very much for your swift response.
> > > We appreciate the opportunity to clarify certain points that may not have been fully explained within the 2500-character limit of the rebuttal:
> > > * **Human evaluation:** We reported intraclass correlation coefficients (ICC) in the paper, which is exactly the inter-annotator agreement metric. ICC is especially appropriate for continuous data like ours, while other metrics have assumptions that do not suit the nature of our ratings, e.g. Cohen’s kappa is designed for categorical data. We additionally transform the scores into rankings to calculate Spearman’s rank correlation coefficient, which stands at 0.437, indicating a moderate positive correlation.
> > > * **NLI-G study:** The absence of ground truth annotations makes it infeasible to separate error-free from erroneous dialogues. As explained in the rebuttal, our conclusion is inferred logically, and the results demonstrate effective evidence retrieval for NLI-G over the Prev setting. We would appreciate your feedback if there are specific aspects of our inference process you believe could be improved.
> > >
> > > We hope these clarifications address your concerns.
> > >
> > > Furthermore, we had prepared more detailed responses to the questions raised in the original review, which we had to extensively trim in the rebuttal due to the character limit. We are ready to provide them now that additional space is available. Please let us know if you would like to review them.
> > >
> > > We look forward to your further suggestions and comments. Thank you.

---

> > > > ### Comment · Reviewer_Z7Mk · 2024-06-04
> > > > **Follow up questions**
> > > >
> > > > *Human evaluation:* Is there more information about the human evaluation? What kind of ratings is the human giving? Is it any number between 0 and 1? Can the paper include these details as well as a citation of prior work where ICC Is applied to measure human agreement?
> > > >
> > > > Please do post the more detailed responses, thank you!

---

> ### Author Response · Authors · 2024-06-06
> **Follow up Response and Detailed Responses**
>
> Thank you very much for your further inquiry and interest in our detailed responses!
> * **Human Annotation:** The annotators were asked to score the consistency and factualness of the current dialogue on a scale from 1 to 10. While several works use ICC to measure annotator agreement, we provide only a few recent references as follows:
>
>     [1] Speech Corpus for Korean Children with Autism Spectrum Disorder: Towards Automatic Assessment Systems (Lee et al., LREC-COLING 2024)
>
>     [2] A Prompt Response to the Demand for Automatic Gender-Neutral Translation (Savoldi et al., EACL 2024)
>
>     [3] Jigsaw Pieces of Meaning: Modeling Discourse Coherence with Informed Negative Sample Synthesis (Singh, EACL Findings 2024)
>
>     [4] FRMT: A Benchmark for Few-Shot Region-Aware Machine Translation (Riley et al., TACL 2023)
>
>     [5] User Simulator Assisted Open-ended Conversational Recommendation System (Zhan et al., NLP4ConvAI 2023)
>
>     [6]  Evaluating the use of large language model in identifying top research questions in gastroenterology. (Lahat et al. Scientific Reports, 2023).
>
>     [7] Inducing Positive Perspectives with Text Reframing (Ziems et al., ACL 2022)
>
>     [8] Listening to Affected Communities to Define Extreme Speech: Dataset and Experiments (Maronikolakis et al., ACL Findings 2022)
>
> 	We will include these details in our paper.
>
> * **More Detailed Responses to the Original Review:**
> ## Prior work discussion
> We will append the related work section with the recommended LLM agent papers.
>
> We especially explain the major differences between our work and the two suggested papers, LEGO (He et al., 2023) and ToM for Multi-Agent Collaboration (Li et al., 2023):
> * **Task:** We address errors in long-term open-domain dialogues between multiple agents. In contrast, Li et al. (2023) tackle the goal-oriented task of defusing bombs in multiple rooms, while He et al. (2023) combine LLM outputs for causality explanation generation, assigning different roles to LLMs to analyze cause and effect, mine knowledge, generate explanations, and refine the explanations.
> * **Approach:**
>     * We develop an error detection and correction framework for each agent’s utterance, while there is no such mechanism in Li et al. (2023). In addition, our task is more challenging. Our agents’ dialogue history and memory contain information about multiple agents, while Li et al. (2023) agents track only task-related information in their belief states, such as bomb locations and colors. Li et al. deal with only 3 agents and 3 colors of bombs in 5 rooms, with clearly defined states to track due to their goal-oriented nature. In contrast, our task involves complex information propagation in open-domain dialogues between 25 agents, each with distinct and varied personal traits. This complexity is particularly challenging given that language models struggle with maintaining self-identity [1]. For instance, the errors illustrated in Figure 1 of our paper are less likely to occur when the agents are discussing bombs.
>
>         [1] Shuster et al., “Am I me or you? State-of-the-art dialogue models cannot maintain an identity.” NAACL-Findings, 2022.
>     * We propose NLI-G and use local models to retrieve evidence from past dialogues for potential errors, whereas He et al. (2023) utilize three additional LLM modules to analyze cause/effect and retrieve world knowledge from Wikipedia.
> ## Elements in Ablation study
> Q2. Memory, history, background, and status constitute the context for utterance generation, originating from GenerativeAgents.
> - Memory: formed by observation and reflection mechanisms, e.g. "People are interested in fostering creativity and innovation."
> - History: previous dialogues of the involved agents
> - Background: agent’s predefined characteristics such as personality, age, profession, current projects, etc. E.g. "Abigail Chen is a digital artist and animator who loves to explore how technology can be used to express ideas."
> - Status: the agent's action before the current dialogue, e.g., "working on her animation project."
>
> We hope we have addressed your concerns. If you have any further questions, please do not hesitate to let us know.

---

### Official Review · Reviewer_cwWx · 2024-05-09

**Rating:** 8
**Confidence:** 4
**Ethics Flag:** 1

**Summary:**

Quality: The paper is in many ways of high quality. The proposed method is fairly sophisticated and makes use of a divide-and-conquer approach to using LLMs, where it is used for many specific tasks in a modular system. The evaluations is pretty thorough and looks into many different aspects of the proposed method.

Clarity: Most things are clear but there is some over-reliance on the Appendix, which should not have to be consulted to understand the paper. (For example, on p.4 instead of referring to the appendix B2, an example should be given in the main text.)

Originality: The approach seems reasonably original, although I do not claim to have a complete overview of the field of quality control of auto-generated dialogue (which is also a quite recent one).

Significance: This is a bit unclear to me. It is not explicitly stated what the ultimate purpouse of the OneDayLife corpus is. Is it intended to be used to train NLP models, for enterntainment, or something else? Since the significance of the overarching project is unclear, so is the significance of the specific work in this paper.

**Questions To Authors:**

- in Figure 1, what do the colours indicate?
- How do you envision making use of the OneDayLife corpus?
- On p.4, why do you assume that inconsistencies are always related to statements referring to a person other than the speaker? Is this a general trait of dialogue, or specific to your corpus?
- On p.4 it says that one purpouse of the graph is to "reduce the negative impact of style discrepancies between pretrained data and raw dialogue utterances". I'm not sure what this refers to, an example would be nice.
- In the caption to figure 4, it's not clear to me why one case is a hallucination and the other is not. It says of the hallucination that "this event could have been entirely fabricated by Ryan, representing a potential harmful hallucination." But is it not possible to check if the event is fabricated, e.g. by comparing to the graph, or using method (b) of asking the referred agent to verify? If the goal is to detect *potential* hallucinations, without checking the truth of the claim, isn't every claim potentually a hallucination?
- In section 2.3, it was not clear to me how the re-generation works. In particular, when an utterance U_c is re-generated as U_c', how do you ensure that subsequent utterances are not rendered inappropriate? For example, if a subsequent utterance refers back to an entity mentioned in U_c but not in U_c'?
- p.6: "Recent works (...) have shown that GPT-4 evaluation correlates more closely with human judgements." - More closely than what? Other LLMs?
- p. 6: "The intraclass correlation coefficients" - is this Fleiss' kappa? In any case, please include the intuitive gloss of the numbers ("high agreement", "low agreement", etc)

Typos:
- p. 2: "a LLM" -> "an LLM" or "a Large Language Model"
- "...determined by whether it is originated from the same speaker or the same dialogue or not." -> "...determined by whether it is originated from the same speaker, the same dialogue, or neither."
- p. 4: "whether the mentioned agent agree with Uc, ..." -> "whether the mentioned agent agree*s* with Uc, along..."
- p. 5 "We develop multiple prompt variances" -> "We develop multiple prompt variants"
- p. 6: "Table 3 shows the ablation study for various prompt design." -> "Table 3 shows the ablation study for various prompt design*s*."
- p. 8 "Randomly pick from structured..." -> "Randomly pick*ing* from structured..."

**Reasons To Accept:**

The paper is mostly well written, and the paper is timely in using a state of the art LLM to generate a dialogue corpus. Only a few papers have been published so far which go in this direction. The proposed methods do indeed seem to improve the quality of the generated corpus.

**Reasons To Reject:**

It is unclear to me what the practical purpouse of the OneDayLife corpus is, but the paper states that it is intended to "mimic complex human behaviours". The goal of the method presented in the paper is to elliminate some undesired behaviours found in the corpus. However, the authors seem to make some implicit assumptions about human dialogue behaviours. For example, that people do not repeat utterances, make false statements, or make inconsistent claims.  Contrary to this, it seems reasonable to assume that most or all of the behaviours that the authors want to elliminate do in fact appear in human-human conversation.  People do at least occasionally repeat themselves, make false statements, and make inconsistent claims. They also exhibit "sudden shifts in opinions, forgetting past statements, and invitations to conflict" (to mention some other behaviours listed in the paper).

Of course, there may be obvious cases of such behaviours in the corpus that would be unexpected or even impossible in human dialogue. But in attempting to remove all "undesired" behaviours, there seems to be a risk of throwing out also human-like repetitions, inconsistencies and false claims as well, thus missing potentially important human behaviours.

To remedy this, it seems you need both an idea of how common these behaviours are in human-human dialoge, as well as an account of when they tend to occur (and when not). Given this information, it would be in principle possible to filter out only the non-humanlike variants of these behaviours, and to evaluate how well one has done this. This, however, would require an effort of looking into the existing literature on human conversational behaviour. If existing research does not provide answers, then ideally one would proceed to collecting such information. This could entail recording and annotating human-human conversations for the phenomena in question. To make such data more precise, it might be a good idea to elicit conversations in a limited setting, to achieve something similar to the OneDayLife corpus (but between humans).

Since a research based account of these behaviours is lacking, a fundamental problem for the authors is that they can only rely on their own intuitions about which dialogue behaviours are human-like.

I do not take this to be a strong argument against accepting the paper. It would, however, be bring this problem to the reader's attention and perhaps discuss it in the future work section.

---

> ### Author Rebuttal · Authors · 2024-05-31
>
> Thank you for your valuable feedback. We appreciate your recognition of our work's quality, clarity, and originality.
>
> ## Significance and Purpose of This Work
> LLMs are widely used in creating dialogue datasets for training NLP models, facilitating decision-making through simulated human interactions, and creating scripts for entertainment. Our study addresses critical issues for these applications.
>
> ## Human Dialogue Assumption
> Your concerns are valid. Yet the highly frequent repetitions, false statements, and inconsistencies in LLM-synthesized dialogues are unrealistic. Prior research generally treats human-annotated dialogues as flawless. Following your advice, we will discuss these challenges in the paper.
>
> ## Response to Questions
>
> 1. In Fig. 1, different colors indicate different agents, with bold-colored phrases identifying the attribute owner or speaker.
> 2. One promising application is the development of LLM-powered Non-Player Characters (NPCs) in games. Please also refer to "Significance and Purpose of This Work."
> 3. We clarify that not all inconsistencies pertain to other agents. NLI-G also detects speaker-specific inconsistencies, as detailed in the "Inconsistency" section on page 4.
> 4. The NLI model is pretrained on single narrative sentences like “Tom did not want to talk about the subject,” which differs from dialogue utterances like “Eddy: I haven't read much Shakespeare.” Therefore, we transform the format to be similar to the pretrained data, e.g. “Eddy hasn't read much Shakespeare.”
> 5. We focus on detecting the “harmful” hallucination -- fabrications misalign with the agent’s role. (p.4)
> 6. We synthesize each dialogue turn by turn and regenerate erroneous utterances immediately instead of correcting one utterance in between the dialogue and leaving the rest as it was.
> 7. GPT-4 evaluation aligns more closely with humans than other automatic metrics and weaker LLMs.
> 8. Intraclass correlation coefficient (ICC) is not Fleiss' kappa; the latter measures the agreement between three or more raters and assumes the ratings are categorical and mutually exclusive, which is unsuitable for our data. ICC is the most ideal metric for our situation since the scores are on an interval scale. We additionally calculate Spearman's rank correlation coefficient by turning the scores to ranking, which is 0.437, indicating a moderate positive correlation.
>
> We will add these clarifications and correct typos in the final version.

---

> > ### Comment · Reviewer_cwWx · 2024-06-04
> >
> > Thank you for your response!

---

### Official Review · Reviewer_1zvS · 2024-05-11

**Rating:** 6
**Confidence:** 4
**Ethics Flag:** 1

**Summary:**

The paper proposes a framework to mitigate (regenerate) inconsistent dialogues in social language multiagent scenarios. The authors first identify issues in the current framework (Park et al), and builds framework components (mostly GPT prompt-based) targeted to mitigate identified issues. Human evaluation with 2 annotators show the framework outperforming the baseline, slightly.

A host of prior work exists (and is cited by the authors) that attempts to improve the dialogues generated by LLM agents in a social multi-agent scenario; this paper's significance lies in extending such prior work to more coherent dialogues across longer periods of "time". It's not entirely clear if this can be solved just by using a longer context-length model or if the authors' targeted modules in the framework are necessary, especially given the not entirely convincing evaluation experiments performed. Nonetheless, the thought put into the NLP engineering side of things is commendable; as such, novelty and significance is reduced.

**Reasons To Accept:**

Good NLP engineering experiment; the authors have put thought into targeting specific identified issues in current social language agent frameworks, and have designed experiments to evaluate performance on these aspects.

**Reasons To Reject:**

It is unclear in the evaluation whether the performance of the authors' proposed system is truly outperforming the baseline- on raw scores, they score exceedingly similar on key metrics like consistency and fluency; no statistical significance is given on the slight improvement. Similarly, the ablation study yields mixed results- without memory, factualness increases? It is unclear how robust the results can be achieved with only 2 annotators on a highly reduced subset of the dataset.

---

> ### Author Rebuttal · Authors · 2024-05-31
>
> Thank you for reviewing and recognizing our paper’s "significance in extending prior work to more coherent dialogues across longer periods of time."
>
> ## Novelty
> Our approach is distinct in developing specific modules to manage problems that are not addressed by merely extending dialogue context lengths. Research [1] has shown the information in the middle of long input contexts can be overlooked.
>
> [1] Lost in the middle: How language models use long contexts (Liu et al., TACL 2024)
> ## Evaluation Robustness
> The fluency is represented by **perplexity**, defined as the exponentiated average negative log-likelihood of text, which **is not a statistical estimate that comes with a known distribution**. Thus, we cannot directly apply statistical tests. Perplexity is a standard evaluation for language models. Lower perplexity indicates a better fit of the model to the data. Besides, "we did not stress on fluency evaluation, as our observations indicate that all generated dialogues are highly fluent" (p.6)
>
> As for consistency scores, the improvements over baselines, while modest, underscore the potential of our framework in a conservative test setup (as described in Appendix C.1 Limitations). However, if we compare the scores pairwisely, we found **ours scored higher than origin in 86.9% of examples**, demonstrating the effectiveness of our framework.
> ## Ablation study yields mixed results:
> As described in Appendix C.1, Our experiment is built on the same context (memory, history, background, and status) from the original dialogue, which may contain inherent problems (repetition, inconsistencies, or factual errors), especially in the last percentile of dialogues. Therefore, removing memory mitigates factual errors stemming from the memory. We expect that these problematic memories will not exist in a complete simulation, as they will reflect the corrected dialogues.
>
> *The agent’s memory is formed by recurrent reflections on their dialogue histories across multiple time points, thus containing many more factual errors than the dialogue history itself. We will add the discussion in the main text.
>
> ## Robustness of human evaluation:
> As mentioned in p.6, we intentionally selected the most challenging subset to rigorously test the model's performance, thereby providing a strong indicator of its robustness. Given our limited budget, this is the best we can do since the human evaluation of the 10% of the full dataset already took 20 hours for each annotator to finish.

---

### Decision · Program_Chairs · 2024-07-10

**Decision:**

Accept

**Comment:**

The paper studies the important problem of LM agents spiralling, hallucinating, and not being factual or consistent. These issues are especially acute in multi-agent simulations due to cascading errors. The framework consists of three steps: screen, diagnose and re-generate. In the screening phase, repetitive utterances and hallucinations are identified. In the diagnosis phase, the flagged utterances are scored through zero-shot prompted LLMs. In the re-generation phase, the flagged utterances are re-generated using a prompted LLM. All reviewers are supportive of acceptance, so the paper is recommended to be accepted subject to the authors incorporating the revisions identified by the reviewers. Additionally, I leave some comments below that need to be addressed at the camera-ready as well. Congratulations!

**Experiments**: While The authors attempt to do a good job at tackling this problem with a “limited budget”, the challenge though is the attempt at building a complicated system in order to tackle all of these different problems at the same time. Unfortunately, this led to doing a poor job on the experimental side - lack of error bars, ablations of partial dataset, evaluation in an offline setting etc. I wish that the paper had just focused on one specific problem and did a more thorough evaluation; including online simulation which is key in this space. I urge the authors to address these issues to the extent possible in the camera ready.

**Related Work to the Proposed Approach**:  Variations of the proposed SDR framework have been proposed in a bunch of domains. Chief among them is Reflexion (https://arxiv.org/abs/2303.11366). Several variations of this have been done for coding and for building coding agents (AlphaCode, SWE-agent etc.) . And in general, several “self-reflection” types of papers have used this in a filter and re-evaluate type of framework. The Cicero Diplomacy paper (https://www.science.org/doi/10.1126/science.ade9097) also proposed a few approaches towards tackling some of these issues. In fact, some pieces of this problem have been studied at least since the Deal-or-No-Deal paper by Mike Lewis from et al in 2017 (https://arxiv.org/abs/1706.05125). Please revise the paper for a better survey of the literature reflecting on the related work.